# A New Approach for the Synthesis of Powder Zinc Oxide and Zinc Borates with Desired Properties

Irina V. Kozerozhets [1,*], Varvara V. Avdeeva [1] , Grigorii A. Buzanov [1], Evgeniy A. Semenov [1], Yulia V. Ioni [1,2] and Sergey P. Gubin [1]

1    Kurnakov Institute of General and Inorganic Chemistry, Russian Academy of Sciences, Leninskii Pr. 31, 119991 Moscow, Russia
2    Moscow Aviation Institute, National Research University, Volokolamskoe sh. 4, 125993 Moscow, Russia
*    Correspondence: irina135714@yandex.ru

**Abstract:** Zinc borates are widely used in industry due to their thermal stability as a flame retardant in the production of plastics, rubber, and other polymer compositions. We have developed a simple and effective approach for the synthesis of zinc borate powders with desired properties, including desired particle size, low bulk density, high reactivity, etc. Zinc borates were prepared by the thermal treatment of a concentrated water–carbohydrate solution of a zinc salt until finely dispersed ZnO was formed, followed by its hydrothermal treatment at 90–300 °C as part of a suspension based on a hot aqueous solution of $H_3BO_3$. According to X-ray powder diffraction, IR spectroscopy, and TG–DSC data, depending on the temperature of hydrothermal treatment, a decrease in the water content in the structure of synthesized zinc borate particles is observed. TEM and SEM data indicate the formation of isometrically shaped zinc borate particles in the nanometer range during hydrothermal treatment above 250 °C. Varying the temperature of the hydrothermal treatment affects the average size and fineness of the structure of the zinc borate particles.

**Keywords:** nanoscale zinc borate; heat treatment; concentrated water–carbohydrate salt solution; hydrothermal treatment; structure improvement; structural water content

## 1. Introduction

Boron chemistry is second only to carbon chemistry in its complicity and variety. Elementary boron tends to form chain, polynuclear, and framework structures, and this property results in the formation of a huge variety of borides [1–4], borates (derivatives of boric acid) [5–7], and hydridoborates (boron cluster anions) [8–10].

Zinc borates are a group of compounds based on Zn:B:O in their various molar ratios [11–13]. Hydrated ($ZnO \bullet B_2O_3 \bullet H_2O$, $2ZnO \bullet 3B_2O_3 \bullet 7H_2O$, $2ZnO \bullet 3B_2O_3 \bullet 3,5H_2O$, etc.) and anhydrous ($ZnO \bullet B_2O_3$, $4ZnO \bullet 3B_2O_3$, etc.) zinc borates are known [12], and are of technological interest as additives to polymer compositions due to their ability to slow down and suppress the processes of smoke formation [14–17]. In the production of paints with high anti-corrosion performance, a joint additive of zinc borate and phosphate bases is used [18,19]. Zinc borate compounds are used as anti-corrosion coatings on machined materials and surfaces [19]. The low toxicity and availability of zinc borates, along with resistance to elevated temperatures and open fire, are valuable properties and determine their widespread use as flame retardants [20], including when creating thermal protective clothing [21]. The dehydration temperature of the hydrated form of zinc borate ($2ZnO \bullet 3B_2O_3 \bullet 3,5H_2O$) is about 290 °C, which gives tangible advantages when mixed with plastics and rubbers at high temperatures (200–300 °C). At the same time, the anhydrous form of zinc borate allows mixing with plastics and rubbers without deteriorating their characteristics with more intense heating [14,17]. Zinc borate is able to interact with the synthetic component of rubbers, with polyvinyl chloride bases, as well as with substances

of the polyurethane foam group, which leads to an increase in the parameters of the oxygen index of polymers [14,17]. In addition, zinc borate plays a role as a fungicide additive that slows down the growth of mold fungi, which is widely used in the manufacturing of building materials [22]. In addition, the preparation of glasses with different properties based on zinc borates is an urgent task of modern industry [23–26].

At present, an increase in interest in the development of methods for the preparation of borate powders with particles of a given size is noted [11,12,15,27]. Thus, during the hydrothermal treatment of nanosized powder CaO and aqueous $H_3BO_3$ at 180 °C for 12 h, submicron priceite ($Ca_2(B_5O_7)(OH)_5 \cdot H_2O$) was synthesized, which has an average particle size of 0.25 μm [27]. Thermal analysis of the synthesized powder showed a weight loss of 16.8 wt% in the region of 25–618 °C and the absence of thermal effects of evaporation of surface water as well as structural water [27]. Subsequent annealing of priceite at 800 °C made it possible to synthesize anhydrous calcium (bis)borate $Ca(BO_2)_2$ and calcium tetraborate $CaB_4O_7$ with an average particle size of 0.35 μm. Thus, annealing up to 800 °C leads to some coarsening of the particles.

It was shown [28] that hydrated zinc borate $2ZnO \bullet 3B_2O_3 \bullet 3,5H_2O$ can be prepared in aqueous suspension when ZnO was allowed to react with an excess of $H_3BO_3$ in the presence of seeds of the final product at elevated temperatures (above 70 °C). This reaction is a widely used industrial process that produces zinc borate powders with an average particle size of about 5 μm. Further heating of the synthesized zinc borate ($2ZnO \bullet 3B_2O_3 \bullet 3,5H_2O$) affords agglomerated anhydrous zinc borate with a particle size of at least 5 μm [28]. Currently, there is no technology that makes it possible to obtain nanosized zinc borate powders with a narrow distribution curve without using heat treatment of the initial precursors at low temperatures.

According to [29–33], the state and forms of water in the structure and on the surface of oxides and oxyhydroxides may differ depending on the particle size. Thus, in [33] it was found that on the surface of submicron particles of silicon oxide and in their volume there is "liquid", "surface-bound" and "molecular-dispersed" water, which has different heats of evaporation. Nanosized powders of oxides and oxyhydroxides are characterized by a higher content of surface-bound water up to 6 wt%, which can be present in the structure of oxides and oxyhydroxides when heated up to 800 °C [30].

Thus, the most urgent issues are both the development of a simple and efficient approach to synthesize zinc borate nanopowders with a narrow curve of particle size distribution, which are of great practical importance, and the study of the state of water on the surface and in the structure of synthesized zinc borate particles.

This work aimed at the development of a new integrated approach for the synthesis of ZnO powder and its use in the preparation of nanopowders of anhydrous zinc borate with controlled properties, as well as the study of the technological conditions of synthesis and features of transformations during heat treatment.

## 2. Results and Discussion

The new approach for the synthesis of powders with desired properties is realized by a multistage thermal treatment of aqueous $Zn(NO_3)_2$ and $D-C_6H_{12}O_6$ at 350 °C for 1 h and at 800 °C for 1 h followed by hydrothermal treatment in a hot aqueous solution of $H_3BO_3$ at different temperatures (Scheme 1).

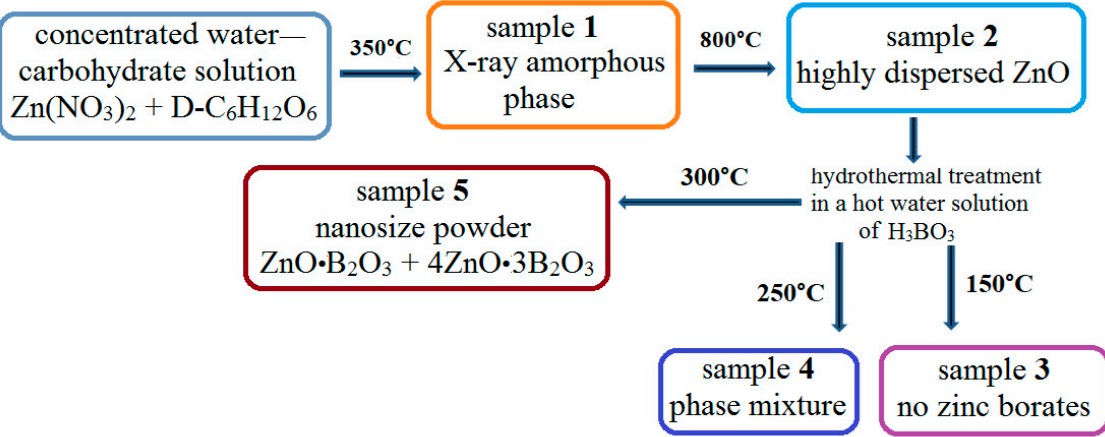

**Scheme 1.** General scheme of a new approach for the synthesis of anhydrous zinc borate nanopowders.

### 2.1. Synthesis of ZnO Powder with Desired Properties (Stage 1)

When crystalline $Zn(NO_3)_2 \bullet 6H_2O$ is allowed to react with a boiling aqueous solution of $D-C_6H_{12}O_6$, zinc aqua-complexes are formed [34], which, upon further concentration of the solution, leads to the isomerization of D-glucose to form D-fructose [34,35], followed by thermal decomposition at 350 °C until the formation of an X-ray amorphous phase containing $Zn^{2+}$ in its structure (sample **1**) (Figure 1). According to the literature data [34–37], thermal treatment of a concentrated water–carbohydrate solution of $Zn(NO_3)_2$ at a temperature of 350 °C leads to the formation of a large amount of volatile organic compounds, such as furans (HMF), levulinic, formic, glycolic, acetic, lactic acids, etc., which, when leaving, form a porous structure of the X-ray amorphous phase with a bulk density of 0.055 $g/cm^3$ (sample **1**). Figure 2 shows SEM images of the synthesized sample **1**, which is a large agglomerated block of particles with a size of 3 μm. Thus, the formation of an X-ray amorphous phase (sample **1**) is an important intermediate step in the formation of zinc oxide powders with desired properties, such as a low bulk density, disordered defect structure, and high reactivity.

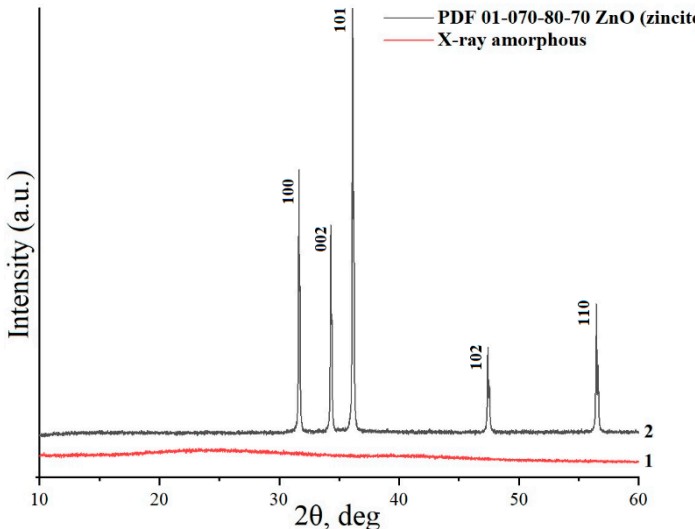

**Figure 1.** X-ray powder diffraction patterns of sample **1** (line 1) and sample **2** (line 2).

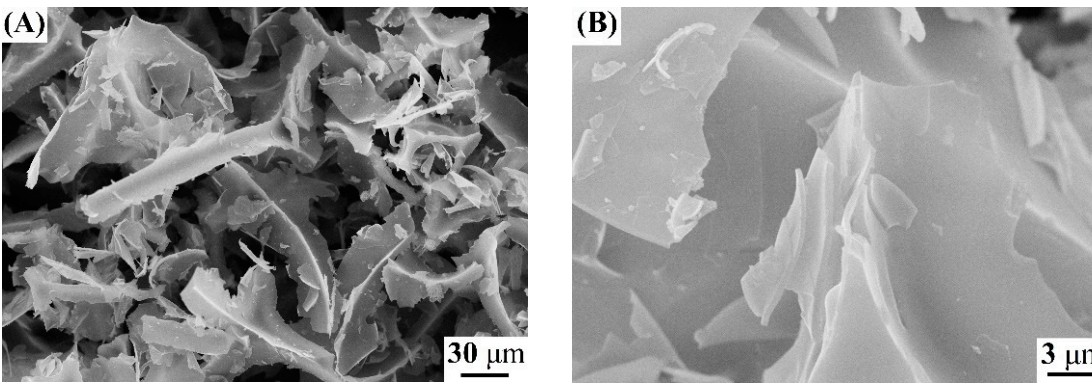

**Figure 2.** SEM images of sample **1** (**A**,**B**).

The annealing of sample **1** at 800 °C leads to the formation of non-agglomerated zinc oxide crystals (sample **2**) corresponding to the PDF 01-070-80-70 phase. There are broadened reflections on the X-ray powder diffraction pattern, indicating the disordered structure of ZnO (Figure 1), which leads to an increase in the reactivity of the compound [31]. Figure 3 shows SEM images of the synthesized non-agglomerated ZnO powder (sample **2**) with an average particle size of 2.3 μm. There are no pronounced thermal effects of dehydration on the DTA/TG curves of sample **2** (Figure 4) and an area of a wide endothermic effect is observed, located in the region of 300–600 °C, which probably corresponds to the decomposition of the X-ray amorphous phase of zinc carbonate. The total weight loss during annealing/heating up to 1000 °C is 7.2 wt%. According to [27,29,30], at the initial stages of heating, the removal of surface-sorbed water molecules is observed.

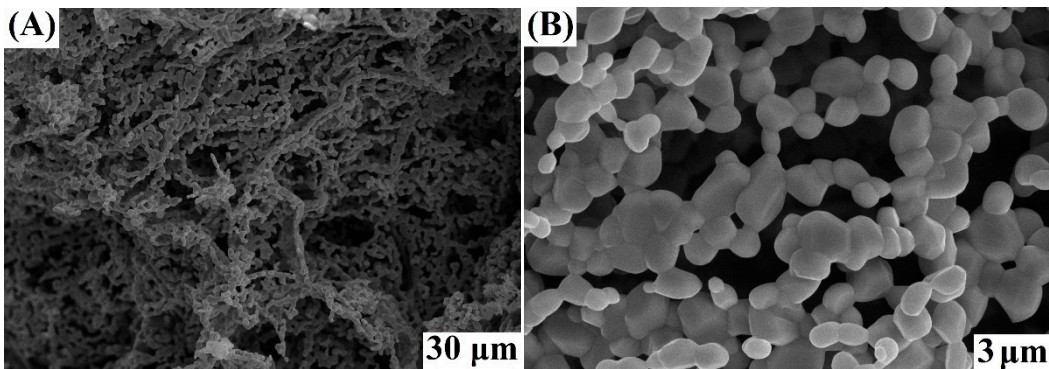

**Figure 3.** SEM images of sample **2** (**A**,**B**).

Table 1 presents the results of studies by the method of CHN-analysis of the content of C, H, N in wt% depending on the processing temperature of the concentrated water–carbohydrate solution of $Zn(NO_3)_3$. The results obtained confirm the data of thermal analysis and show a decrease in the carbon content in the sample with an increase in the heating temperature up to the values of the sensitivity limit of the measurement method.

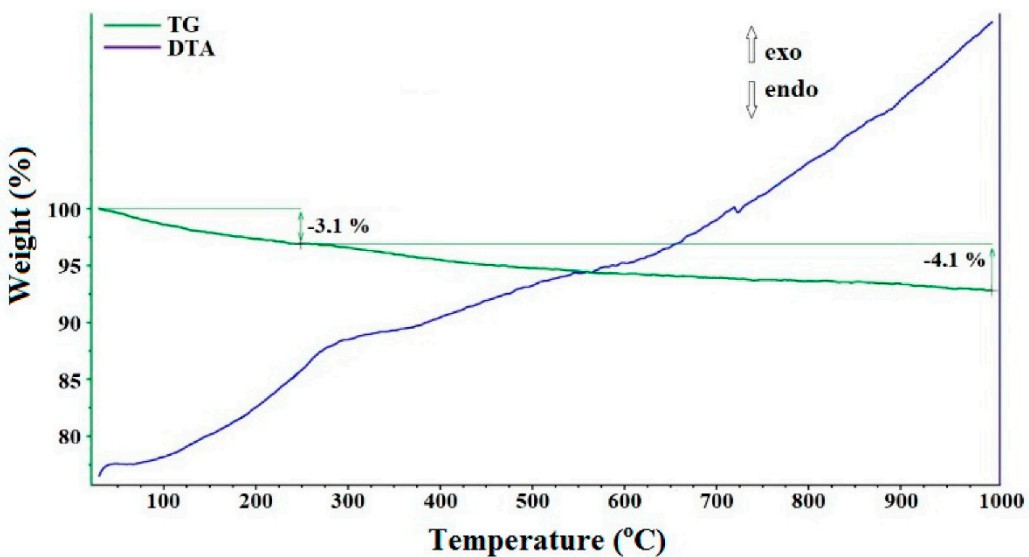

**Figure 4.** DTA and TG curves for sample **2**.

**Table 1.** Analysis of the content of C, H, N (wt%) depending on the processing temperature of the concentrated water–carbohydrate solution of $Zn(NO_3)_3$.

| | Processing Temperature, °C | | | | | | | | |
|---|---|---|---|---|---|---|---|---|---|
| | **150** | **250** | **350** | **450** | **550** | **650** | **750** | **800** | **850** |
| C | 22.85 | 37.69 | 34.71 | 8.87 | 5.64 | 1.1 | 0.25 | - | - |
| H | 5.1 | 2.68 | 2.71 | 2.43 | 2.31 | 0.43 | 0.21 | 0.14 | - |
| N | 5.63 | 2.36 | 2.25 | 0.82 | - | - | - | - | - |

The properties of the synthesized ZnO powders are presented in Table 2.

**Table 2.** Properties of synthesized ZnO powders and anhydrous zinc borate.

| | Sample 2 | Sample 5 |
|---|---|---|
| Particle size range | 0.1–7.2 μm | 10–90 nm |
| Average particle size | 2.3 μm | 43 nm |
| Thermal conductivity coefficient W/(m K) | 0.07 | 0.15 |
| Specific surface area ($m^2$/g) | 7 | 30 |
| Bulk density (g/$cm^3$) | 0.4 | 1.2 |

The use of finely dispersed ZnO is widely represented in modern industry. First of all, finely dispersed ZnO has the ability to absorb a wide range of electromagnetic radiation, including UV, IR, microwave and radio frequency ranges. The absorption of UV radiation formed the basis for the creation of various products in medicine (creams, masks, dressings, etc.), in the construction industry (varnishes, paints, glass, polymer fibers), etc. The use of finely dispersed ZnO to prepare components for semiconductor sensors, devices, solar cells, and UV filters is based on the properties of ZnO as a transparent wide-gap semiconductor.

Thus, a new approach to synthesize oxide powders (by the example of ZnO) with controlled properties has been developed and tested: a low bulk density from 0.4 g/$cm^3$ and a low thermal conductivity from 0.07 W/(m K). The absence of agglomeration, the absence of faceting of the obtained particles, and the disordered defective structure of the synthesized oxides determine the high reactivity of the substance, and in particular, the low

temperatures of phase transformations and solid-phase processes. Weak aggregation makes it easy to carry out the process of further modification in order to obtain new materials based on the studied oxides.

### 2.2. Synthesis of Finely Dispersed Zinc Borate Nanopowder (Stage 2)

A series of experiments were carried out on the hydrothermal treatment of sample **2** (ZnO) in a suspension obtained on the basis of an aqueous solution of $H_3BO_3$ at temperatures of 150, 250 or 350 °C.

It is known that when dissolved, $B_2O_3$ reacts with water to form orthoboric acid, which exhibits very weak acidic properties because of the addition of an $OH^-$ ion due to the electron pair of oxygen and the free orbital of boron [38]. Intensive mixing of hot aqueous $H_3BO_3$ and ZnO at 90 °C in a water bath for 30 min allows a uniform distribution of components to be achieved.

According to the X-ray powder diffraction results, hydrothermal treatment of the suspension obtained when sample **1** reacted with aqueous $H_3BO_3$ at 150 °C ($\tau$ = 72 h) does not lead to a change in the phase composition of the sample. The pattern (Figure 5) contains reflections corresponding to the phase PDF 01-070-80-70 zincite (ZnO) and PDF 01-078-04-70 $B(OH)_3$ sassolite (sample **3**). The X-ray powder diffraction results are in agreement with the IR spectroscopy data. Figure 6 shows the IR spectrum of sample **3**. The band at 1190 cm$^{-1}$ corresponds to H–O–H bending vibrations, the band at 1409 cm$^{-1}$ is assigned to B–O stretching vibrations, and the band at 3200 cm$^{-1}$ corresponds to stretching vibrations of the OH groups in $B(OH)_3$. These data agree with those reported in [39–42]. A band at 465 cm$^{-1}$ corresponds to the ZnO stretching vibrations in zinc borate [43].

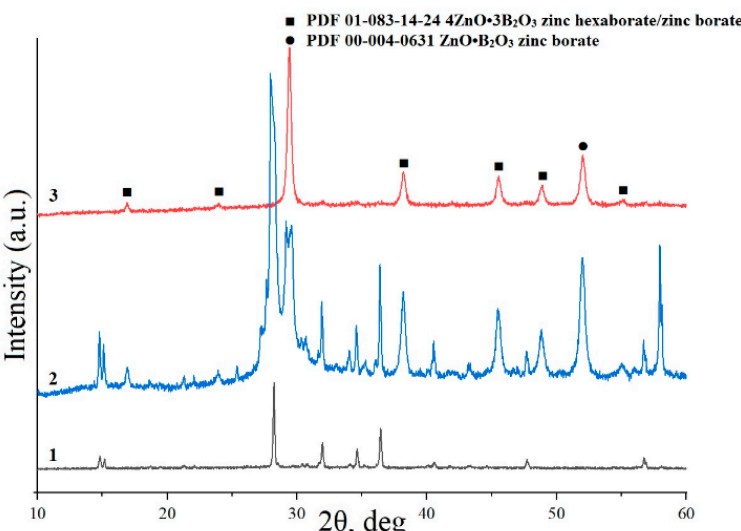

**Figure 5.** X-ray powder diffraction patterns of samples **3** (curve 3), **4** (curve 2), and **5** (curve 3).

Thus, the obtained IR spectroscopy data confirms that sample **3** contains bands characteristic of both boric acid and zinc borate.

According to the X-ray powder diffraction (Figure 5), hydrothermal treatment of the suspension obtained when sample **1** was allowed to react with aqueous $H_3BO_3$ at 250 °C ($\tau$ = 72 h) leads to the formation of sample **4**, consisting of phases of zinc borates $ZnO \bullet B_2O_3$ and $4ZnO \bullet 3B_2O_3$ (PDF 01-083-14-24 and PDF 00-004-06-31), zincite (ZnO) (PDF 01-070-80-70) and $B(OH)_3$ sassolite (PDF 01-078-04-70), which is reflected in the IR spectrum (Figure 6). The IR spectrum of sample 4 is similar to the spectrum of sample **3**, but there is some redistribution of bands in the region of 1500–400 cm$^{-1}$. Broad bands of the OH stretching vibrations (3200 cm$^{-1}$) and B–O stretching vibrations (1425 cm$^{-1}$) are observed. A band corresponding to H–O–H bending vibrations shifts to 1194 cm$^{-1}$. In the DTA/TG curves of sample **4** (Figure 7) in the range of 25–215 °C, there are low-temperature

endothermic effects at 121 °C, 142 °C, and 163 °C with a total weight loss of 18.2 wt%, which correspond to the decomposition of $B(OH)_3$ residues in sample **4**. The endothermic effect in the range of 890–984 °C with a maximum at 960 °C corresponds to the melting point of zinc borate [12].

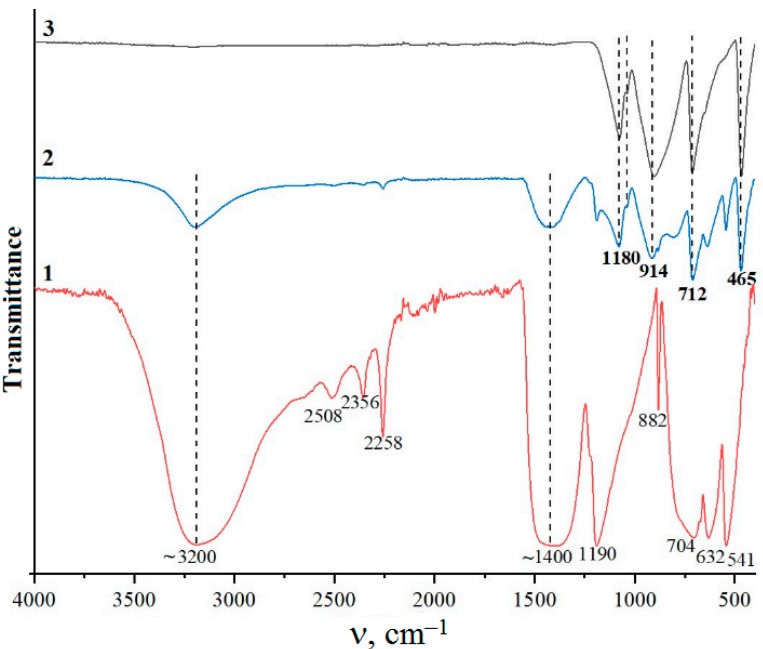

**Figure 6.** IR spectra of samples **3** (curve 1), **4** (curve 2), and **5** (curve 3).

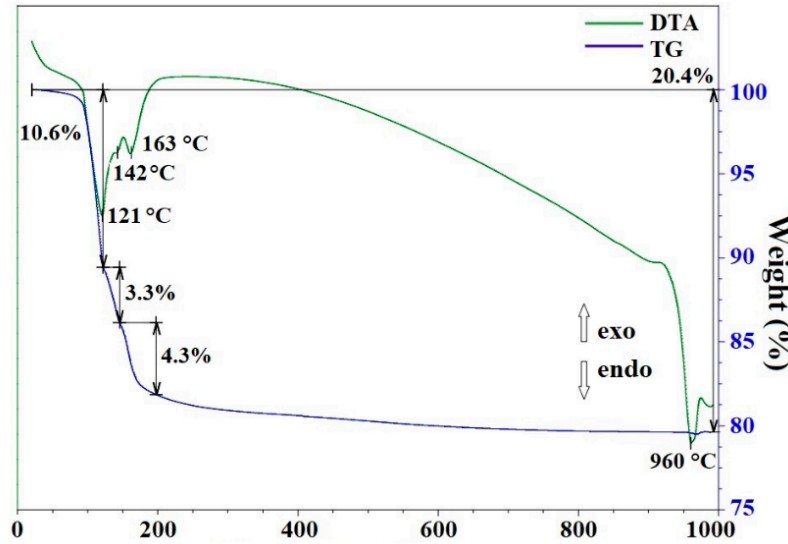

**Figure 7.** DTA and TG data for sample **4**.

Hydrothermal treatment of the suspension obtained when sample **1** was allowed to react with aqueous $H_3BO_3$ at 300 °C for 72 h, according to the X-ray powder diffraction data (Figure 5), leads to the formation of sample **5**, consisting of phases of zinc borates $ZnO \bullet B_2O_3$ and $4ZnO \bullet 3B_2O_3$ (PDF 01-083-14-24 and PDF 00-004-06-31). In the IR spectrum (Figure 6), there are bands characteristic of anhydrous zinc borate. Bands of the OH stretching vibrations (3200 cm$^{-1}$) and H–O–H bending vibrations (1190 cm$^{-1}$) are absent, confirming that the sample is water-free; a broad band of B–O stretching vibrations from boric acid present in the IR spectra of samples **1** and **2** at 1425 cm$^{-1}$ is also absent. At the same time, a band of the Zn–O stretching vibrations is observed at 465 cm$^{-1}$. On the DTA/TG curves (Figure 8) of sample **5** in the region of 900–1000 °C, an endothermic effect is recorded with a maximum at 969 °C, corresponding to the melting point of zinc borate. In this case, the total weight loss of the sample is about 2.2 wt% with noticeable breaks on the TG curve, which indicates the removal of non-localized OH groups from the sample structure. SEM data (Figure 9A,B) make it possible to conclude that the synthesized samples of zinc borate are spherical agglomerates with an average size of 1 μm to 30 μm, while each spherical agglomerate consists of nanosized particles of zinc borate (Figure 9B–E) with an average size of 43 nm and a narrow distribution curve. The properties of the synthesized zinc borate powder are presented in Table 2.

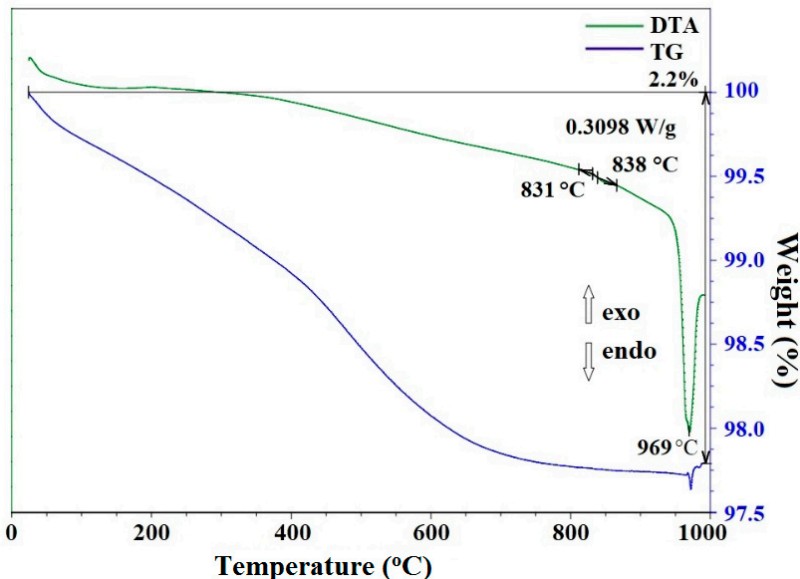

**Figure 8.** DTA and TG data for sample **5**.

The synthesized nanosized powder of anhydrous zinc borate can be widely used as an antiperspirant in the production of plastics and rubbers, in the production of products and coatings with high anticorrosion performance, etc. The promising potential of using the synthesized zinc borate as a seed for the growth of zinc borate crystals with given properties, which will reduce the economic component of the production of hydrated and anhydrous zinc borate powders, should be noted.

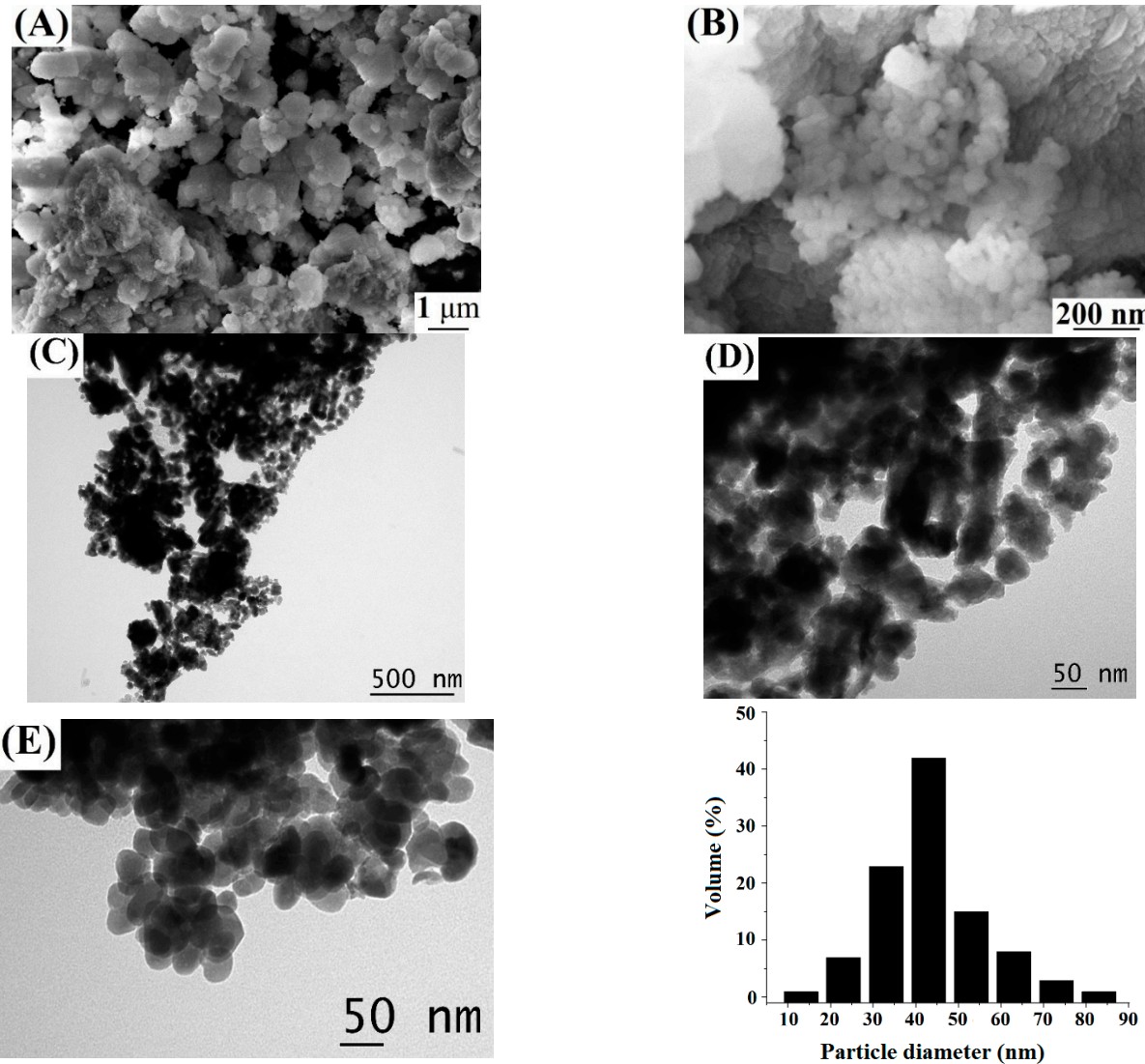

**Figure 9.** SEM (**A**,**B**) and TEM (**C**–**E**) images, and sample particle size distribution curve for sample **5**.

## 3. Materials and Methods

### 3.1. Synthesis

Here, we used two successive stages to perform the present study.

Stage 1. Synthesis of powder ZnO with desired properties.

The method for obtaining finely dispersed ZnO powder is based on the technology discussed in detail [44,45]. Crystalline $Zn(NO_3)_2 \bullet 6H_2O$ (85 g, special quality grade (Russian State Standard), Unikhim, Russia) was added in portions with vigorous stirring to a boiling aqueous solution of carbohydrate obtained by dissolving 50 g of D-glucose in 200 mL of distilled water until a brown viscous substance formed. Subsequent heating at a heating rate of 5 °C/min with holding at temperatures of 350 °C and 800 °C for 1 h made it possible to obtain finely dispersed nanosized ZnO powder.

Stage 2. Synthesis of finely dispersed zinc borate nanopowder.

The finely dispersed nanopowder ZnO (1.62 g) synthesized in Stage 1 was kept at 90 °C in a water bath in a hot aqueous solution of $H_3BO_3$ (3.71 g of $B_2O_3$ was dissolved in 10 mL of $H_2O$) with vigorous stirring (300 rpm) for 30 min. We used $B_2O_3$ of special quality grade (Russian State Standard; OJC "Boron", Russia). The resulting suspension was placed in a titanium liner of the autoclave, which was placed in a preheated SNOL electric furnace at 150–300 °C. The filling factor of the autoclave was 20%. The exposure time of the autoclave in the electric furnace was 72 h. After holding at the specified

period of time, the autoclave was removed from the furnace, cooled with running water, and then depressurized. The prepared sample was removed from the titanium insert, then washed with distilled water, and kept in an oven at $85 \pm 5$ °C for 12 h to remove surface-sorbed water.

In Table 3, the preparation methods and the phase compositions of the samples are compared.

**Table 3.** Methods for preparation and the phase composition of samples.

| | Method for Preparation | Phase Composition |
|---|---|---|
| sample **1** | heating of aqueous $Zn(NO_3)_2$ and D-glucose at 350 °C ($\tau = 1$ h) | X-ray amorphous |
| sample **2** | heating of sample **1** at 800 °C ($\tau = 1$ h) | ZnO |
| sample **3** | hydrothermal treatment of the suspension obtained when sample **1** reacted with aqueous $H_3BO_3$ at 150 °C ($\tau = 72$ h) | ZnO, $H_3BO_3$ |
| sample **4** | hydrothermal treatment of the suspension obtained when sample **1** reacted with aqueous $H_3BO_3$ at 250 °C ($\tau = 72$ h) | ZnO, $H_3BO_3$, $ZnO \bullet B_2O_3$, $4ZnO \bullet 3B_2O_3$ |
| sample **5** | hydrothermal treatment of the suspension obtained when sample **1** reacted with aqueous $H_3BO_3$ at 300 °C ($\tau = 72$ h) | $ZnO \bullet B_2O_3$, $4ZnO \bullet 3B_2O_3$ |

### 3.2. Methods

Elemental analysis was performed on a C, H, N-analyzer EA1108 from Carlo Ebra Instruments (Italy). Combustion of the synthesized samples was provided by adding $Co_2O_3$ to the capsule. A sample weighing up to 1 mg was burned in automatic mode in the reaction tube of the analyzer at T = 980 °C with a pulsed oxygen supply. The combustion products entered the lower part of the reaction tube connected to the oxidation catalyst and reduced copper. Products were analyzed using a thermal conductivity detector with computer processing of the obtained chromatographic data.

IR spectra were measured on a Bruker Alpha IR Fourier spectrometer with a Platinum ATR attachment; measurements were performed in the range of 400–4000 cm$^{-1}$; accumulation mode, 48 scans.

The morphology and average particle size of samples were determined by electron microscopy. Scanning electron microscopy images were obtained using a CAMSCAN-S2 microscope and a Carl Zeiss Supra 40 scanning electron microscope. For the CAMSCAN-S2 microscope, the study was carried out at an accelerating voltage of 20 kV (secondary electron detection mode; focal length, 10 mm). For the Carl Zeiss Supra 40 microscope, the samples were placed on a holder which was placed inside a chamber with a ~$10^{-6}$ mbar vacuum. The accelerating voltage was 1–10 kV; the aperture was equal to 30 μm. For transmission electron microscopy (TEM) studies, we used a JEOL Jem-1011 instrument and measurements were performed at an accelerating voltage of 80 kV. The samples were deposited on copper grids coated with carbon films by ultrasonic sputtering.

X-ray powder diffraction studies of samples were performed using a Bruker Advance D8 diffractometer (radiation, $CuK_\alpha$; $2\theta = 10$–$60°$; scanning step, $0.0133°$; plexiglass cuvettes). The values of interplanar distances and intensities of maxima in the patterns of samples were compared with the ICDD PDF-2 database in order to identify crystalline phases.

Specific surface area, specific pore volume, as well as porosity were determined using low-temperature nitrogen adsorption on an ATX-06 analyzer (KATAKON, Novosibirsk, Russia). Before measurements, samples were degassed in a nitrogen flow at 150 °C for 35 min.

Differential scanning calorimetry studies were performed using a SDT Q600 combined TGA/DTA/DSC thermal analyzer at a heating rate of 10 °C/min in the range of 20–1000 °C using $Al_2O_3$ as a reference. The studies were carried out in corundum crucibles. The weight of sample **2** was 13.2440 mg; the weight of sample **4** was 12.4550 mg; the weight of sample **5** was 17.5470 mg.

Bulk density was measured by the pycnometric method with a measurement error of 10 wt%.

Thermal conductivity was measured on an ITP-MG4 Zond SKB instrument by the probe method.

## 4. Conclusions

We have developed a simple and effective method for obtaining industrially important zinc borate powders, which are agglomerates of nanoparticles with an average size of 43 nm and a narrow distribution curve. This method consists of the primary synthesis of finely dispersed ZnO from a concentrated water–carbohydrate solution of zinc nitrate, its suspension in an aqueous solution of $H_3BO_3$, and subsequent hydrothermal treatment at 90–300 °C. It was shown that the formation of the target product, anhydrous zinc borate, which is of great technological importance as a fire retardant and fungicide in the composition of polymer compositions, is possible during hydrothermal treatment of the suspension obtained when sample **1** reacted with aqueous $H_3BO_3$ at 300 °C ($\tau$ = 72 h). TEM and SEM methods have proven the formation of isometric zinc borate nanoparticles during hydrothermal treatment above 250 °C. Due to the synthesis parameters, the developed approach has significant prospects for implementation in industry.

**Author Contributions:** Investigation, I.V.K. and Y.V.I.; writing—original draft preparation, I.V.K.; writing—review and editing, V.V.A.; formal analysis, G.A.B. and E.A.S.; conceptualization, S.P.G.; funding acquisition, Y.V.I. All authors have read and agreed to the published version of the manuscript.

**Funding:** This work was supported by the Russian Science Foundation (project no. 22-19-00110).

**Acknowledgments:** Analytical research was performed using equipment at the shared facility center of the National Research Center "Kurchatov Institute"—IREA which operates under financial support provided by the Ministry of Science and Higher Education of the Russian Federation, agreement no. 075-11-2021-070 dd. 19 August 2021.

**Conflicts of Interest:** The authors declare no conflict of interest.

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
