# Peer review of "A New Approach for the Synthesis of Powder Zinc Oxide and Zinc Borates with Desired Properties"

_inorganics, doi:10.3390/inorganics10110212_

Round 1

Reviewer 1 Report

In this work, a new approach for the synthesis of powder zinc oxide and zinc borates with desired properties: particle size, low bulk density, and high reactivity was proposed. It's a nice work and contains some interesting results. In addition, this paper is well written and well organized. While I think the manuscript has the following question that should be solved:

1.       How to define the approach proposed in this work is a new one.

2.       Whether the purity or target compound can be controlled 

Author Response

Reviewer #1

In this work, a new approach for the synthesis of powder zinc oxide and zinc borates with desired properties: particle size, low bulk density, and high reactivity was proposed. It's a nice work and contains some interesting results. In addition, this paper is well written and well organized. While I think the manuscript has the following question that should be solved:

  1. How to define the approach proposed in this work is a new one.

Answer: The novelty of the approach proposed in the work lies in 2 aspects: (i) a new method for the synthesis of finely dispersed ZnO powder and (ii) its approbation for the synthesis of nanosized zinc borate powders  (ZnО•B2O3, 4ZnО•3B2O3).

  1. Whether the purity or target compound can be controlled

Answer: The purity of finely dispersed ZnO is determined by the purity of the feedstock and the control of the content of carbon, hydrogen, and nitrogen at all stages of heat treatment (Table 2). The yield of the target product is determined by the conditions of hydrothermal treatment. Thus, hydrothermal treatment of ZnO in an H3BO3 solution at 300C makes it possible to obtain a mixture of zinc borate ZnO•B2O3 (4 wt.%), 4ZnO•3B2O3 (96 wt.%). Thus, the content of the target product of the anhydrous zinc borate has been achieved and is determined by the temperature of the hydrothermal treatment.

Dear reviewers,

thank you for your time and your interest in our work. Your fruitful comments allow us to improve the quality of the manuscript.

Reviewer 2 Report

The paper itself is fine as far as concerning a preparation method of oxides.  However, there are no scientific approaches in the paper to demonstrate that the presented synthetic methodology is actually useful in various applications. 

Author Response

Reviewer #2

1.The paper itself is fine as far as concerning a preparation method of oxides.  However, there are no scientific approaches in the paper to demonstrate that the presented synthetic methodology is actually useful in various applications.

Answer: The methodology presented in this work for obtaining powders with desired properties may be of interest for industrial applications, both as a methodology for the synthesis of ZnO powders with a narrow distribution curve, and as a methodology for the synthesis of anhydrous zinc borates. It should be noted that the phases of anhydrous zinc borate were obtained in the nanoscale range, which allows them to be used as seeds for the preparation of aqueous phases of zinc borates.

Dear reviewers,

thank you for your time and your interest in our work. Your fruitful comments allow us to improve the quality of the manuscript.

Reviewer 3 Report

The authors of the manuscript proposed a new approach for the synthesis of zinc borate nanopowders in two stages. It consists of synthesizing ZnO powder following its hydrothermal treatment at 90–300 ℃ as part of a suspension based on a hot aqueous solution of B2O3. The proposed method has a scientific interest but some moments raise questions.

1) According to table 3, the average particle size of sample 2 (ZnO powder) is 2,3 μm. So, to denote that the ZnO powder is nanosized or highly dispersed is incorrect and contradicts the experimental data. More than that, I can't agree with the authors that the XRD lines of the ZnO phase are broad in Fig.1-2 (lines 180-182). Try to determine the crystallite sizes by the broadening of diffraction lines.

2) Line 143: “Combustion of graphene oxide films…” What has the graphene oxide to do with it?

3) Line 212: What method was used to measure thermal conductivity?

4) Line 220: I guess sample 2 (not sample 1) was used to synthesize zinc borate nanopowder. Please, check it.

5) Fig. 9. If we talk about nanosized powder the distribution from 10 to 90 nm is quite wide, not narrow.

6) The authors proposed the method for the synthesis of zinc borate nanopowders with desired properties: particle size, low bulk density, high reactivity, etc. Bulk density and reactivity depend on particle size. So, one important question remains. How particle size of ZnO and zinc borate can regulate by the proposed method?

Although the study is quite interesting and many research techniques have been employed to characterize the material, I would be able to recommend this manuscript for further publication only after minor revision.

Author Response

Reviewer #3

1. According to table 3, the average particle size of sample 2 (ZnO powder) is 2,3 μ So, to denote that the ZnO powder is nanosized or highly dispersed is incorrect and contradicts the experimental data. More than that, I can't agree with the authors that the XRD lines of the ZnO phase are broad in Fig.1-2 (lines 180-182). Try to determine the crystallite sizes by the broadening of diffraction lines.

Answer: We fully agree with the remark about nanoscale, the text has been amended.

Regarding the remark about the dispersion of the synthesized ZnO powder, it should be noted that ZnO is characterized by a bulk density equal to 0.4 g/cm3 due to the formation of a large number of pores in the synthesized powder (the bulk density for commercial ZnO is 5.61 g/cm3), so the use of the definition finely dispersed in this work we consider competent. Calculation by the Debye-Scherrer method according to the diffraction pattern (Fig. 1-2) made it possible to obtain an average particle size of 1.9 μm.

We used “finely dispersed” term throughout the whole manuscript.

2. Line 143: “Combustion of graphene oxide films…” What has the graphene oxide to do with it?

Answer: This is our mistake, corrected, thank you.

3. Line 212: What method was used to measure thermal conductivity?

Answer: Added to the text. Thermal conductivity was measured on an ITP-MG4 Zond SKB instrument by the probe method.

4. Line 220: I guess sample 2 (not sample 1) was used to synthesize zinc borate nanopowder. Please, check it.

Answer: Corrected, thank you.

5. Fig. 9. If we talk about nanosized powder the distribution from 10 to 90 nm is quite wide, not narrow.

Answer: Corrected. It was assumed in the work that as can be seen in Figure 9, the content of particles in the range from 30 to 70 nm is 89%, so the wording “narrow curve for particle size distribution” was chosen.

6. The authors proposed the method for the synthesis of zinc borate nanopowders with desired properties: particle size, low bulk density, high reactivity, etc. Bulk density and reactivity depend on particle size. So, one important question remains. How particle size of ZnO and zinc borate can regulate by the proposed method?

Answer: The presented method makes it possible to control the particle size, their agglomeration, etc. by selecting the optimal synthesis parameters, which are indicated in the work. So, with an increase in the concentration of D-glucose in a solution of more than 25 wt %, agglomeration of ZnO is observed, which is indirectly evidenced by an increase in the bulk density of ZnO. With a decrease in the concentration of D-glucose in a solution of less than 20 wt %, the average particle size of ZnO approaches 10 μm, which also does not correspond to the task. The selected concentration range is optimal for ZnO synthesis. The particle sizes of zinc borates directly depend on ZnO and hydrothermal treatment parameters, including the autoclave holding time in the oven.

Dear reviewers,

thank you for your time and your interest in our work. Your fruitful comments allow us to improve the quality of the manuscript.

Reviewer 4 Report

The main objective of the article is the synthesis of zinc borate nanopowders with controlled properties.

The article presents a new method for the preparation of these materials. However, to be considered in industrial processes, information must be provided, for example, on the performance at each stage and globally. It also presents some points that need to be treated in greater depth.

In the DTA shown in Figure 4, the peak at 300 °C and why it occurs should be mentioned.

If I understand correctly, the sample used to do the CHN analysis tests shows a decrease in the amount of carbon with heat treatment. Therefore, it should correspond to the loss of TG shown in Figure 4. The explanation must be deepened and not only say that the decrease is due to the loss of water.

Figure 8 does not explain the change in the observed mass (approximately 3%), which corresponds to a change in the DTA baseline. What is this process due to?

minor comments

In the experimental section indicate the make of the Jem-1011 equipment, for example JEOL.

For thermal analysis, indicate the atmosphere and sample size and what type of crucible was used.

In figure 4, the indication exo and endo must be included in the graphics or in the text. And the DTA units too.

In figure 6 the y-axis must be indicated (transmittance)

In figure 7 the indication exo and endo must be included. And the DTA units too.

Author Response

Reviewer #4

The main objective of the article is the synthesis of zinc borate nanopowders with controlled properties.

The article presents a new method for the preparation of these materials. However, to be considered in industrial processes, information must be provided, for example, on the performance at each stage and globally. It also presents some points that need to be treated in greater depth.

  1. In the DTA shown in Figure 4, the peak at 300 °C and why it occurs should be mentioned.

Answer. Corrections have been made to the text. The DTA and TG curves show an area of a wide endothermic effect located in the region of 300-600ºС, which probably corresponds to the decomposition of the X-ray amorphous phase of zinc carbonate, the content of which does not affect the synthesis of the target product - phases of zinc borates.

  1. If I understand correctly, the sample used to do the CHN analysis tests shows a decrease in the amount of carbon with heat treatment. Therefore, it should correspond to the loss of TG shown in Figure 4. The explanation must be deepened and not only say that the decrease is due to the loss of water.

Answer: For thermogravimetric studies, sample 2 was used, which is ZnO obtained after heating at 800 for 1 hour (Table 1). According to Table 2, it does not contain C content, within the error of the measuring equipment.

  1. Figure 8 does not explain the change in the observed mass (approximately 3%), which corresponds to a change in the DTA baseline. What is this process due to?

Answer: On the DTA and TG curves (Fig. 8) of sample 5 in the region of 900-1000, an endothermic effect is recorded with a maximum at 969, corresponding to the melting of zinc borate. In this case, the total mass loss of the sample is about 2.2 wt % with noticeable breaks on the TG curve, which indicates the removal of non-localized OH groups from the sample structure.

minor comments

In the experimental section indicate the make of the Jem-1011 equipment, for example JEOL.

Answer: Corrected, thank you.

For thermal analysis, indicate the atmosphere and sample size and what type of crucible was used.

Answer: Corrected, thank you.

In figure 4, the indication exo and endo must be included in the graphics or in the text. And the DTA units too.

Answer: Corrected, thank you.

In figure 6 the y-axis must be indicated (transmittance)

Answer: Corrected, thank you.

In figure 7 the indication exo and endo must be included. And the DTA units too.

Answer: Corrected, thank you.

Dear reviewers,

thank you for your time and your interest in our work. Your fruitful comments allow us to improve the quality of the manuscript.

Round 2

Reviewer 4 Report

No comments